# Attenuation of cGAS/STING activity during mitosis

Brittany L Uhlorn[1], Eduardo R Gamez[2], Shuaizhi Li[3], Samuel K Campos[1,3,4,5]

**The innate immune system recognizes cytosolic DNA associated with microbial infections and cellular stress via the cGAS/STING pathway, leading to activation of phospho-IRF3 and downstream IFN-I and senescence responses. To prevent hyperactivation, cGAS/STING is presumed to be nonresponsive to chromosomal self-DNA during open mitosis, although specific regulatory mechanisms are lacking. Given a role for the Golgi in STING activation, we investigated the state of the cGAS/STING pathway in interphase cells with artificially vesiculated Golgi and in cells arrested in mitosis. We find that whereas cGAS activity is impaired through interaction with mitotic chromosomes, Golgi integrity has little effect on the enzyme's production of cGAMP. In contrast, STING activation in response to either foreign DNA (cGAS-dependent) or exogenous cGAMP is impaired by a vesiculated Golgi. Overall, our data suggest a secondary means for cells to limit potentially harmful cGAS/STING responses during open mitosis via natural Golgi vesiculation.**

## Introduction

Cells possess intrinsic sensory pathways as part of the innate immune system to detect microbial infection or other physiological insults (1). Foreign nucleic acids are often recognized as pathogen-associated molecular patterns through a number of pattern recognition receptors (2), causing activation of NFκB-dependent inflammatory cytokine responses and/or IRF3/7-dependent type-I interferon (IFN-I) responses (1, 3). The cGAS/STING pathway is recognized as a central component of innate immunity for cytosolic DNA recognition and downstream IFN-I responses (4, 5, 6, 7, 8). Cytosolic DNA is recognized by the enzyme cGAS, triggering production of the cyclic dinucleotide 2′,3′-cGAMP (9). STING, a transmembrane ER protein (10, 11), is activated by direct binding to cGAMP (12).

Upon activation by cGAMP at the ER, dimeric STING undergoes a conformational change (13) and traffics to the Golgi, a prerequisite for assembly of the STING/TBK1/IRF3 complex and downstream IFN-I responses (14). cGAMP-dependent STING recruitment of TBK1 (15) can lead to phosphorylation of IRF3 and NFκB, stimulating both IFN-I and proinflammatory cytokine responses (10, 16, 17). Trafficking

of STING to the Golgi is regulated by several host factors, including iRHOM2-recruited TRAPβ (18), TMED2 (19), STIM1 (20), TMEM203 (21), and ATG9A (22). STING activation at the Golgi requires palmitoylation (23) and ubiquitylation (24, 25), allowing for assembly of oligomeric STING and recruitment of TBK1 and IRF3 (26, 27, 28). STING also interacts with the ER adaptor SCAP at the Golgi to facilitate recruitment of IRF3 (29). In addition to innate defense against microbial infections, cGAS/STING is involved in cellular responses to DNA damage and replicative/mitotic stress (5, 30, 31, 32, 33, 34, 35, 36). DNA damage, replicative stress, chromosomal instability, and mitotic errors can lead to the formation of micronuclei which can trigger antiproliferative IFN-I and senescence responses via cGAS/STING (37).

Unabated activation of cGAS/STING can lead to harmful auto-inflammatory and senescence responses, exemplified by type-I interferonopathies associated with mutations in STING (38, 39, 40, 41) or mutations in the DNases TREX1 and DNASE2 that normally clear cells of cGAS-stimulatory DNA (42, 43, 44, 45). Given the harmful effects of cGAS/STING hyperactivation, cells need regulatory mechanisms to avoid self-stimulation of cGAS/STING during mitosis. Cytosolic compartmentalization of cGAS was initially proposed as a mechanism, but nuclear chromosomes and cytosolic compartments mix upon mitotic nuclear envelope breakdown (NEBD), suggesting a more elaborate means of cGAS/STING attenuation during cell division.

The chromatinized nature of cellular genomic DNA has been proposed to mitigate cGAS/STING activity, with histones structurally marking DNA as "self." This is an attractive model as many DNA viruses sensed by cGAS/STING upon initial entry, trafficking, and uncoating (before viral DNA replication) contain either naked, unchromatinized dsDNA (e.g., herpesviridae (46, 47)) or DNA that is packaged with non-histone viral core proteins (e.g., adenoviridae, poxviridae, and asfarviridae (48, 49, 50, 51, 52)).

Because cGAS localizes to condensed chromosomes upon NEBD (35, 53), others have asked whether cGAS is activated by chromosomes, and if not, what mechanisms exist to prevent such self-activation. Recent studies have revealed that i) chromosome-bound cGAS is tightly tethered to chromatin, potentially via interactions with H2a/H2b dimers, ii) chromatin interaction does not involve the DNA-binding domains of cGAS required for "typical" activation by dsDNA, and iii) that chromosome binding results only in weak activation of cGAS with relatively low production of cGAMP (34, 53, 54).

[1]Cancer Biology Graduate Interdisciplinary Program, The University of Arizona, Tucson, AZ, USA  [2]Department of Physiology, The University of Arizona, Tucson, AZ, USA  [3]Department of Immunobiology, The University of Arizona, Tucson, AZ, USA  [4]BIO5 Institute, The University of Arizona, Tucson, AZ, USA  [5]Department of Molecular and Cellular Biology, The University of Arizona, Tucson, AZ, USA

Correspondence: skcampos@email.arizona.edu

Given the importance of the Golgi in STING-dependent activation of IRF3, we hypothesize a parallel mechanism for cGAS/STING regulation during open mitosis–Golgi vesiculation (55, 56). Here, we find that chemical dispersal of the Golgi abrogates cGAS/STING-dependent phospho-IRF3 responses to transfected DNA. Furthermore, we show that cGAS/STING activity in response to transfected DNA is diminished during open mitosis, correlating with the vesiculated state of the mitotic Golgi. This Golgi-dependent weakening of cGAS/STING responses to transfected DNA occurs at the level of STING, as cGAS activity is down-regulated upon mitotic chromosome binding but largely unaffected by Golgi integrity. The vesiculated state of the mitotic Golgi may, therefore, provide an additional safeguard mechanism, ensuring that potentially harmful cGAS/STING responses to self-DNA are minimized during cell division.

# Results and Discussion

### Human keratinocytes respond to foreign DNA via cGAS/STING

We chose to investigate activity of the cGAS/STING pathway in HaCaTs, a spontaneously immortalized human keratinocyte cell line (57). This cell line is a good model for primary keratinocytes that serve a barrier function and as host cells for a number of viral infections, including papillomaviruses, herpesviruses, mosquito-transmitted alphaviruses and flaviviruses (58, 59, 60, 61, 62, 63, 64), and numerous bacterial and fungal pathogens (65, 66). Although prior work has shown that HaCaT cells express an intact cGAS/STING pathway (60, 67, 68, 69, 70, 71), we sought to ensure the cGAS/STING pathway was functional and responsive in our HaCaT line and that this pathway was the primary mode of IRF3 phosphorylation in response to foreign DNA.

HaCaTs were transfected with siRNAs targeting the cGAS/STING pathway and subsequently transfected with endotoxin-free dsDNA plasmid pGL3 (Fig 1). IRF3 was phosphorylated (pIRF3) in response to DNA transfection, exemplifying the ability of exogenous dsDNA to activate the cGAS/STING pathway in HaCaTs. IRF3 phosphorylation was impaired when pGL3 was transfected after siRNA knockdown of

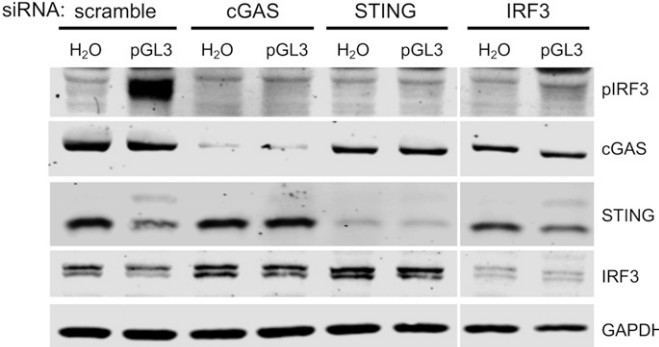

**Figure 1. Human keratinocytes respond to foreign DNA via cGAS/STING.**
HaCaTs were transfected with siRNAs for 16 h, followed by transfection with 500 ng pGL3 or water for 90 min. Lysates were analyzed for cGAS/STING component knockdown and pathway activity by SDS–PAGE and Western blot. Source data are available for this figure.

cGAS, STING, or IRF3, confirming that HaCaT cells use the cGAS/STING pathway as the primary mechanism of activating a pIRF3 response to foreign DNA.

### Golgi vesiculation but not fragmentation impairs cGAS/STING activity at the level of STING

Activated STING traffics to the Golgi to oligomerize and complex with TBK1/IRF3. Because Golgi trafficking is critical for STING/TBK1/IRF3 complex assembly and activation, we hypothesized that a vesiculated Golgi would prevent cGAS/STING from responding to foreign DNA. To test this idea, we used the Golgi-disrupting compounds nocodazole (NOC), golgicide A (GCA), and brefeldin A (BFA). At high dose, NOC treatment causes reversible Golgi fragmentation and redistribution to ER-exit sites (ERES) (72), whereas GCA and BFA treatment cause more drastic Golgi vesiculation and dispersal by targeting the Arf1 guanine nucleotide exchange factor GBF1 (73, 74). Indeed, NOC treatment of HaCaT cells caused a pronounced fragmentation of the Golgi as seen by immunofluorescence (IF) microscopy for trans-Golgi markers p230 and TGN46, whereas GCA and BFA treatment completely vesiculated the Golgi apparatus (Fig 2A and B), mimicking the dispersed mitotic Golgi. As expected, transfection of pGL3 resulted in clustering of STING with p230- and TGN46-positive Golgi membranes and nuclear import of IRF3, indicative of pathway activation (Fig 2A–D). Addition of GCA or BFA completely abrogated activation of STING and IRF3, whereas NOC had minimal effect, suggesting that whereas a fragmented Golgi can support pGL3-stimulated STING/IRF3 activation, a dispersed Golgi cannot (Fig 2A–D). Likewise, IRF3 phosphorylation was induced upon transfection of HSV-60 oligonucleotide DNA, calf-thymus DNA (CTD), or pGL3 plasmid in HaCaTs with intact or fragmented Golgi; however, GCA- or BFA-mediated Golgi vesiculation impaired DNA-dependent IRF3 phosphorylation (Fig 3A), in agreement with prior literature (75).

To determine if the requirement for intact Golgi was specific for cGAS/STING or involved a more broad inhibition of IRF3 phosphorylation, we investigated activation of the RIG-I–like receptors (RLRs). The dsRNA-mimic polyinosinic-polycytidylic acid (pIC) stimulates RLR-family members which recognize intracellular viral RNA products through the mitochondria-resident adaptor protein MAVS to elicit NFκB inflammatory and IRF3/7-dependent IFN-I responses (76, 77). pGL3-dependent pSTING, pTBK1, and pIRF3 responses were abolished by Golgi dispersal (Fig 3B and C). In contrast, transfection of pIC elicited pTBK1 and pIRF3 responses regardless of GCA treatment (Fig 3B and D), suggesting that the cGAS/STING pathway, but not the RLR pathway, is regulated by Golgi morphology. Exogenous stimulation of HaCaT cells with the STING ligand cGAMP was sensitive to GCA-mediated Golgi disruption (Fig 3E), and GCA had no effect on cGAMP production in response to pGL3 transfection (Fig 3F), indicating that Golgi dispersal blocked the pathway downstream of cGAS.

The GCA-induced repression of cGAS/STING activation was also reflected when measuring pGL3-dependent downstream transcriptional responses. RT-qPCR at 4 and 8 h post-pGL3 transfection revealed that induction of IFNB1 (Fig 4A), the ISGs Viperin, IFI6, HERC5, IFIT2, and IFIT3 (Fig 4B–F), and chemokines CXCL10 and CXCL11 (Fig 4G and H) was significantly dampened in the presence of

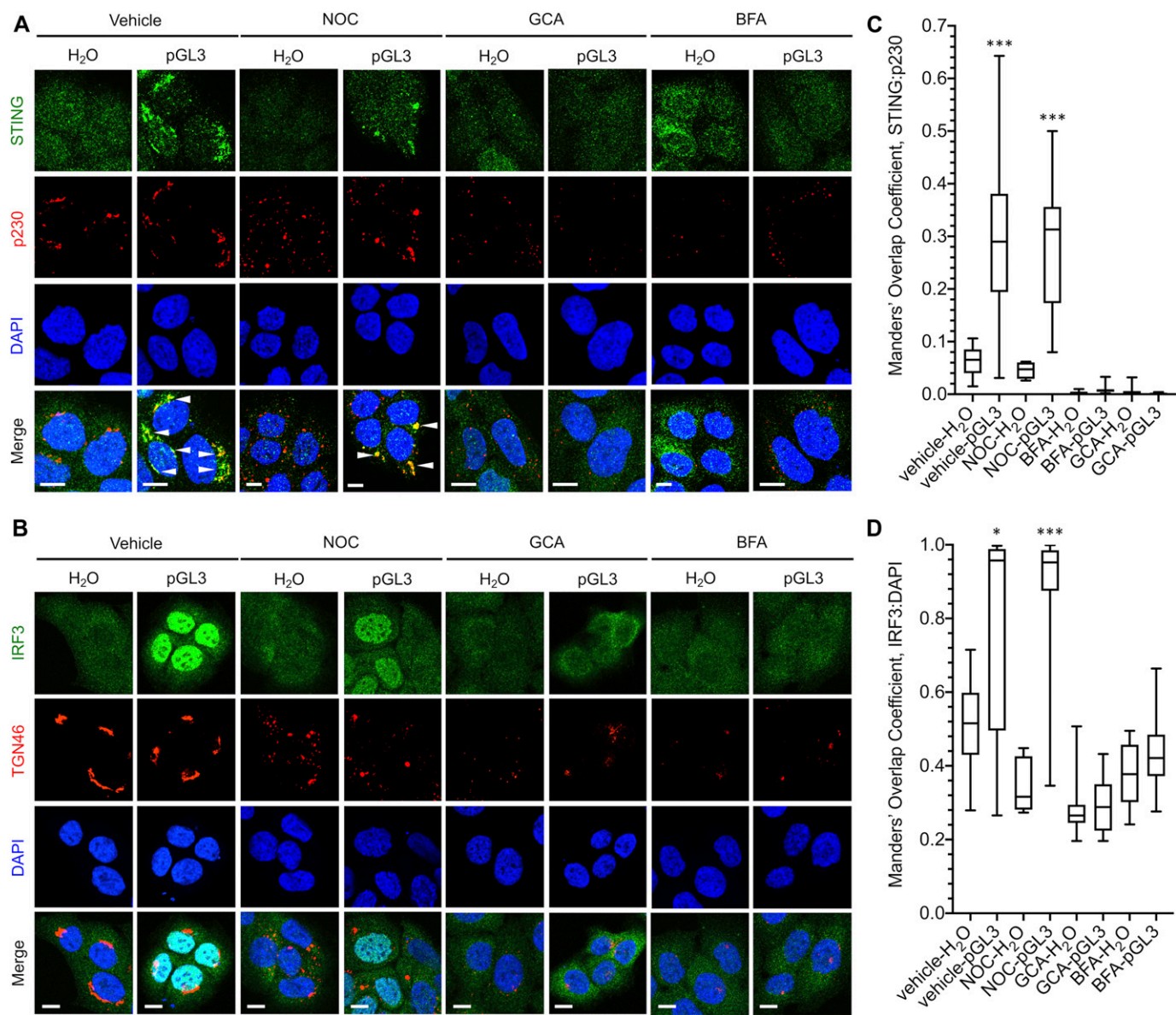

**Figure 2. Effects of Golgi disruption on DNA-dependent subcellular localization of STING and IRF3.**
HaCaT cells were pretreated with vehicle, NOC, GCA, or BFA before a 90-min transfection with $H_2O$ or pGL3. **(A, B)** Cells were fixed and stained for (A) STING and p230 or (B) IRF3 and TGN46 before DAPI staining. **(A, B)** Representative micrographs are shown in (A, B), white arrowheads indicate overlap. **(C, D)** Manders' overlap coefficients from multiple micrographs were plotted for (C) STING:p230 and (D) IRF3:DAPI. *$P < 0.01$, ***$P < 0.0001$. Scale bars = 10 $\mu m$.

GCA. Overall, these data show that cGAS/STING activity is blunted at the level of STING upon Golgi vesiculation, similar to what may occur during mitosis.

## cGAS/STING activity is attenuated during mitosis

We next investigated the impact of natural mitotic Golgi vesiculation on the ability of cGAS/STING to sense and respond to exogenous DNA. Secretory ER to Golgi traffic is blocked during mitosis (78, 79, 80, 81). Golgi integrity is dependent on cargo transport from ERES, and mitotic arrest of COPII-dependent ERES traffic causes Golgi dispersal (82, 83). As assembly of the activated STING/TBK1/

IRF3 complex requires STING transport from ERES to the Golgi (14), we hypothesized that mitotic Golgi dispersal and inactivation of ERES would blunt cGAS/STING responses.

We devised a method to synchronize cells at prometaphase (Fig 5A). Briefly, cells were cultured at 100% confluence for 48 h in 1% serum, leading to quiescent arrest in $G_0$. Cells were released by replating in 10% serum, allowing for $G_1$ reentry and progression to S phase. At 24 h post-$G_0$ release, cells were synchronized at prometaphase with low-dose NOC for 12 h. Upon NOC washout, synchronized cells progressed through mitosis, returning to $G_1$ within 3 h. Propidium iodide (PI) staining showed cells enriched at $G_2/M$ after NOC treatment, and the most cells back in $G_1$ by 3 h post-NOC

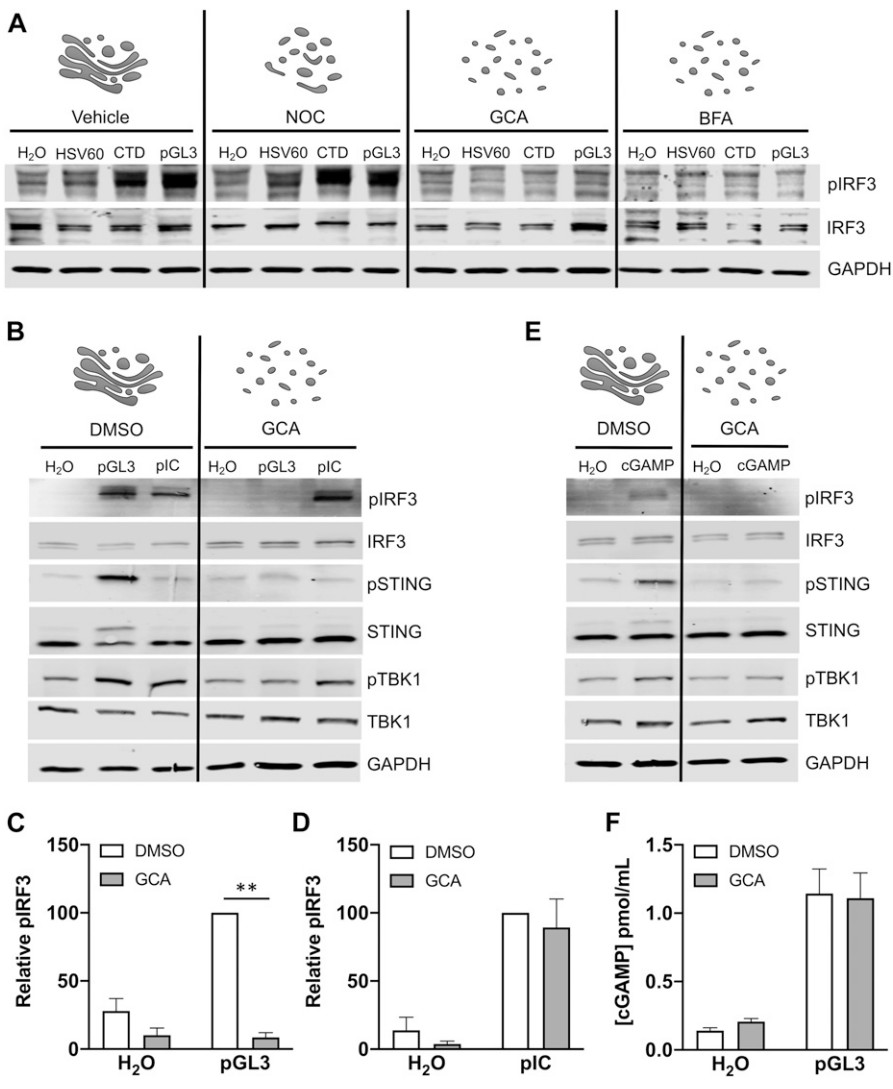

**Figure 3. Effects of Golgi disruption on cGAS/STING activity.**
**(A)** Cells were treated with vehicle, nocodazole (NOC), golgicide A (GCA), or brefeldin A (BFA) for 1 h prior and during a 90-min transfection with HSV-60 oligonucleotide, calf-thymus DNA (CTD), or plasmid pGL3. **(B, C, D)** Transfection of vehicle- or GCA-treated cells with pGL3 or pIC, and (C, D) densitometric quantification of pIRF3 blots. **$P < 0.001$, n = 5 biological replicates. **(E)** Vehicle- or GCA-treated cells were treated with $H_2O$ or 12.5 μg cGAMP for 90 min before SDS–PAGE and Western blot for cGAS/STING pathway components. **(F)** cGAMP production in vehicle- and GCA-treated cells upon pGL3 transfection. n = 3 biological replicates, with n = 2 technical replicates each.
Source data are available for this figure.

release (Fig 5A). Phosphorylated histone H3 (pH3) was enriched in NOC-synchronized cells, decreasing as cells returned to $G_1$ (Fig 5B).

We used this scheme to assess mitotic cGAS/STING responses to pGL3 transfection. Cells were transfected while in the prometaphase after the 12 h NOC sync (non-released group, NR), or at 0, 60, 120, or 180 min post-release, and cGAS/STING activity was assessed 90 min post-transfection. Cells arrested at the prometaphase mounted a very weak pIRF3 response to pGL3 transfection compared with cells which had returned to $G_1$ after a 3 h release (Fig 5B, compare lanes 6–10). These arrested cells had condensed chromosomes with dispersed Golgi (Fig 5D, NR). STING had a cytosolic but granular distribution in arrested cells, which did not change upon pGL3 transfection. In contrast, STING clearly localized to p230-positive Golgi structures upon pGL3 transfection of $G_1$ cells (Fig 5D and E). On average, across four independent synchronization experiments, pGL3-dependent phosphorylation of IRF3 was attenuated during mitosis, only becoming robust at 180 min (Fig 5C), when the bulk population of cells reached $G_1$ and had intact Golgi with clear STING localization upon pGL3 transfection (Fig 5D and E).

## cGAS and STING are nonresponsive during mitosis

Recent work has shown that upon NEBD, cGAS localizes to mitotic chromosomes via direct binding to H2a/H2b dimers (53). However, this binding is not via the DNA-binding domains of cGAS that underlie DNA-dependent activation, resulting in a relatively low production of cGAMP. Thus, chromatin appears to blunt cGAS activation (34, 53), suggesting a means for the cell to avoid cGAS-driven IFN responses to self-chromosomes during open mitosis. One unexplored aspect is whether chromatin-bound cGAS, or the pool of cGAS that might remain cytosolic during open mitosis, would still be responsive to foreign (naked) DNA—and whether that activation would then cause IRF3 phosphorylation.

We observed only low activation of IRF3 in response to pGL3 transfection in NOC-arrested prometaphase cells (Figs 5B and C and 6A). To determine if the dampening of the pathway in these arrested cells was at the level of cGAS or STING, we measured cGAMP production in response to pGL3 transfection and examined cGAS subcellular distribution. Similar to pIRF3, cGAMP production was

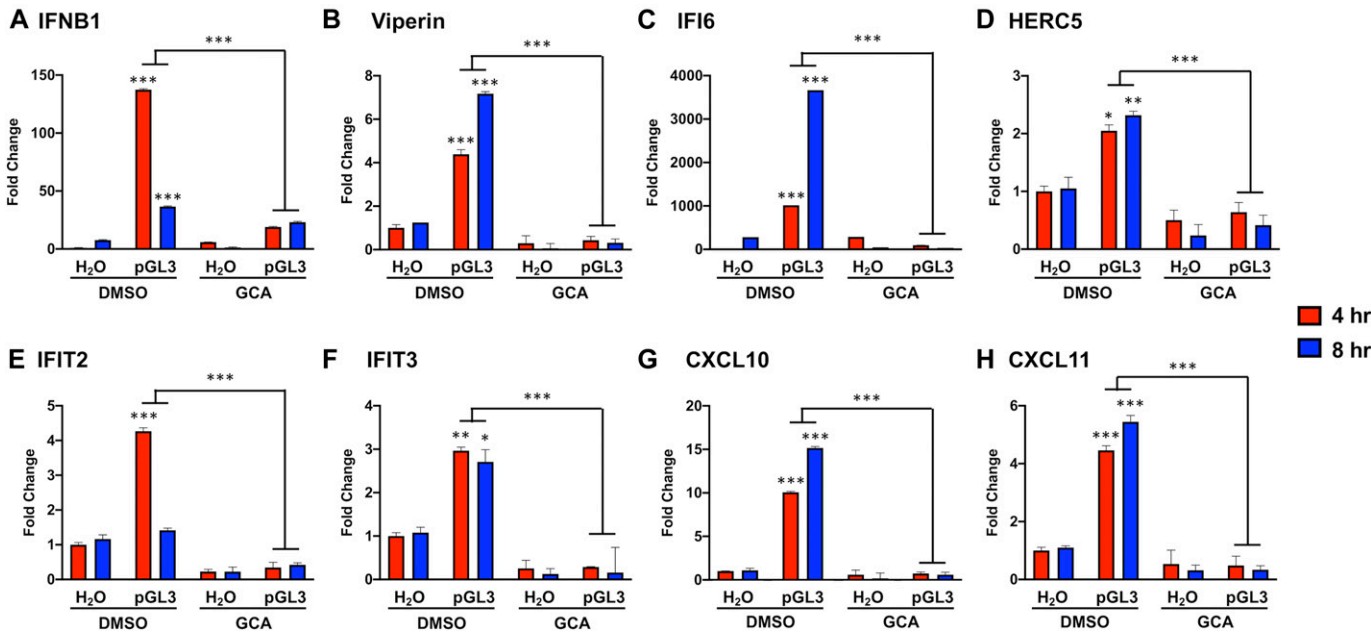

**Figure 4. Golgi vesiculation impairs DNA-dependent IFN, ISG, and chemokine gene transcription.**
HaCaT cells were treated with GCA or vehicle for 1 h before a 90-min pGL3 transfection. **(A, B, C, D, E, F, G, H)** Transcript levels of (A) IFNB1, (B, C, D, E, F) ISGs, and (G, H) chemokine genes were measured via RT-qPCR and normalized to TATA-binding protein. *$P < 0.01$, **$P < 0.001$, ***$P < 0.0001$, for comparisons of 4 and 8 h DMSO-treated pGL3 groups to 4 and 8 h DMSO-treated $H_2O$ controls and for 4 and 8 h DMSO-treated pGL3 groups to 4 and 8 h GCA-treated pGL3 groups, n = 3 technical replicates.

blunted in transfected arrested cells (NR) compared with asynchronous interphase cells (Fig 6B). Confocal microscopy of asynchronous cells revealed that cGAS was mostly nuclear, although some signal was evident within the cytosol (Fig 6C), in agreement with a recent report (54). Within arrested cells, the vast majority of cGAS was chromatin bound and Golgi were well dispersed (Fig 6D). These results agree with recent work showing that cGAS is predominantly nuclear, regardless of cell cycle phase or activation status (54). Combined, our data suggest that chromatin-bound cGAS is unable to produce a robust cGAMP response to either chromosomes or transfected DNA.

Although we detected slightly elevated levels of cGAMP in unstimulated arrested versus asynchronous cells (Fig 6B), the subtle difference was not significant, consistent with recent work showing cGAS generates only low levels of cGAMP upon activation by chromatin (34, 53). Mitotic cells could potentially take up exogenous cGAMP via SLC19A1 (84, 85) or LRRC8 (86) transporters, or directly from neighboring cells (87, 88). Recent work has revealed a primordial role for cGAMP-dependent STING activation in triggering autophagy through WIPI2/ATG5 in a TBK1/IRF3-independent manner (75). These Golgi-derived autophagosomes promote clearance of cytosolic DNA and incoming DNA viruses such as HSV1 (75). Although mitotic cells appear to avoid classical induction of autophagy via CDK1 phosphorylation of ATG13, ULK1, and ATG14 (89, 90), downstream STING-dependent activation of autophagophore formation/elongation via WIPI2/ATG5 during open mitosis could be deleterious to daughter cell survival (91, 92).

Considering the potential for activation of STING by low levels of endogenous cGAMP or uptake of exogenous cGAMP, we assessed whether mitotic Golgi vesiculation would prevent STING activation by exogenous cGAMP. Neither transfection of pGL3 nor addition of exogenous cGAMP stimulated pIRF3 or pSTING in arrested cells (Fig 6E), suggesting that Golgi vesiculation reinforces cGAS inactivation as a secondary barrier to cGAS/STING activation during open mitosis. During revision of this manuscript, an article was published describing how cGAS is phosphorylated and inactivated by the kinase Cdk1-cyclin B during mitosis (93). Using different cell types, their findings largely agree with what we observe herein—that chromosome-bound cGAS is inactive to stimulation by exogenous transfected DNA in mitosis and that STING is nonresponsive to exogenous cGAMP in mitotic cells. Interestingly, when mitotic cGAS inactivation was blunted by the use of the Cdk1 inhibitor RO-3306, despite increased cGAMP levels or addition of exogenous cGAMP, there was still lack of activated pSTING and pIRF3 (93), consistent with our data suggesting pathway inhibition by a vesiculated Golgi. Fragmentation and vesiculation of the Golgi is a natural $G_2/M$ checkpoint for mitotic progression (55, 94, 95, 96), thus we are unable to experimentally prevent Golgi dispersal during mitosis to determine if cGAMP-dependent STING activity would then be restored.

Given that cGAS/STING-dependent elevation of pIRF3 during mitosis (particularly during prolonged mitosis) can induce apoptosis (53) and activated STING can induce a potentially harmful autophagic response (75), a parallel dampening mechanism like Golgi dispersal likely serves an important role to limit potentially harmful cGAS/STING signaling during open mitosis. Furthermore, many other viruses and bacteria induce dramatic alteration of Golgi membranes during infection (97, 98, 99, 100, 101). Microbial alteration of Golgi integrity may be an important unrecognized tactic to blunt host cGAS/STING responses.

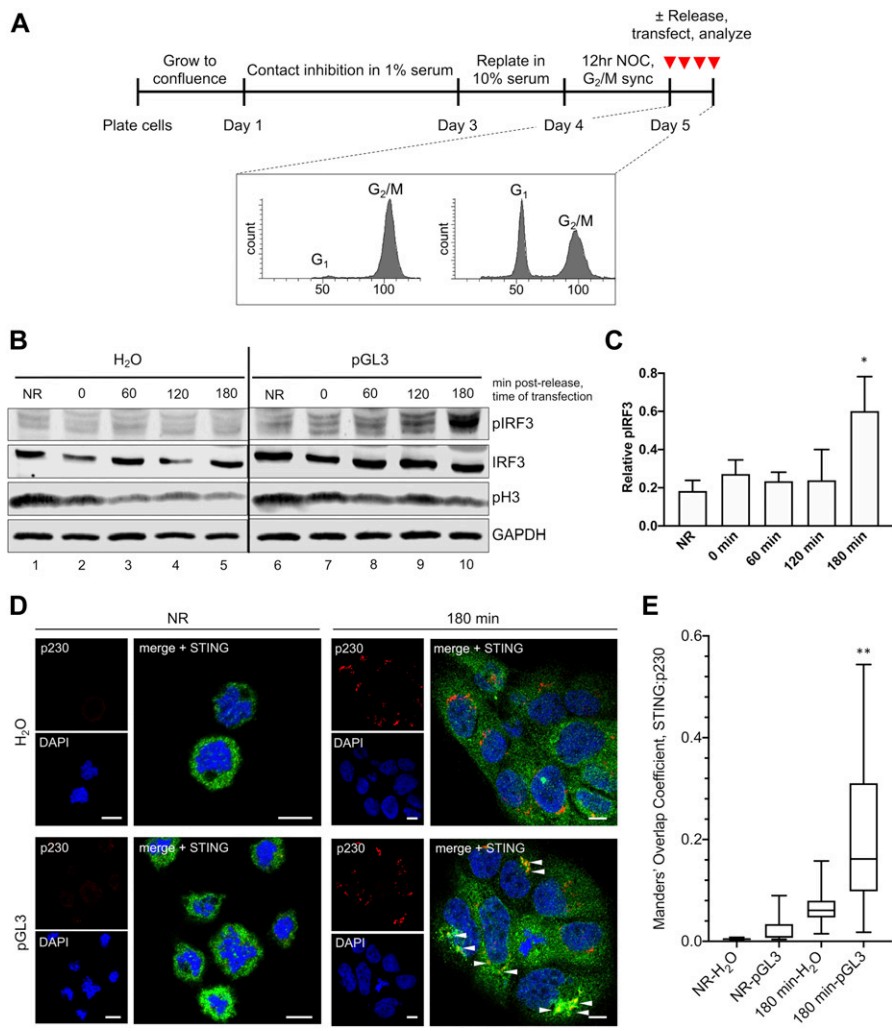

**Figure 5. cGAS/STING activity is attenuated during mitosis.**
**(A)** Cell synchronization method. Cells were pre-synchronized in $G_1$ by contact inhibition and growth in low serum. Cells were then released by replating at subconfluence in 10% serum and synchronized in prometaphase with NOC. Cell cycle analysis by propidium iodide staining. Cells were transfected with pGL3 for 90 min at various times post-release from NOC. **(B)** pIRF3 activation in response to pGL3 was dampened when arrested cells were transfected, gradually returning as cells completed mitosis and returned to interphase. **(C)** Densitometric quantification of relative pIRF3 increase in response to pGL3 transfection across multiple blots. *$P < 0.05$, for comparison of 180 min to NR groups, n = 4 independent biological replicates. **(D)** Golgi localization of STING was impaired when arrested cells (NR) were transfected, but was restored upon return to interphase (180 min). **(E)** Manders' overlap coefficients from multiple micrographs were plotted for STING:p230, **$P < 0.001$, scale bars = 10 $\mu$m.

# Materials and Methods

## Tissue culture

HaCaT cells were grown in high-glucose DMEM (11965-092; Gibco) supplemented with 10% FBS (A31606-02; Gibco) and antibiotic-antimycotic (15240062; Thermo Fisher Scientific). Cells were cultured at 37°C with 5% $CO_2$ and passaged every 2–3 d to maintain subconfluence.

## Nucleic acid transfections

HaCaTs were plated at 60,000 cells per well in a 24-well plate. Cells were transfected with 500 ng dsDNA oligonucleotide HSV-60 (tlrl-hsv-60n, 60 bp; InvivoGen), calf-thymus DNA (D4764, >20 kb; Sigma-Aldrich), or endotoxin-free pGL3 (E1751, 5.3 kb; Promega), or 500 ng high molecular weight poly(I:C) (tlrl-pic, 1.5–8 kb; InvivoGen) using Lipofectamine 2000 (11668; Thermo Fisher Scientific) in OptiMEM (Life Technologies). At various time points post-transfection, cells were washed once with PBS and lysed in 1× RIPA lysis buffer (50 mM

Tris–HCl, pH 8.0, 150 mM NaCl, 1% NP40, 0.5% sodium deoxycholate, and 0.1% SDS), supplemented with 1× reducing SDS–PAGE loading buffer (62.5 mM Tris, pH 6.8, 10% glycerol, 2% SDS, 0.5% bromo-phenol blue, and 5% $\beta$-mercaptoethanol), 1× protease inhibitor cocktail (P1860; Sigma-Aldrich), 1 mM PMSF, and 1× PhosSTOP phosphatase inhibitor cocktail (04906845001; Roche). Samples were boiled for 5 min at 95°C and stored at –80°C until separated and analyzed by SDS–PAGE.

## siRNA experiments

Pooled scramble (sc-37007), cGAS (sc-95512), STING (sc-92042), and IRF3 (sc-35710) siRNA duplexes were obtained from Santa Cruz Biotechnologies. HaCaTs were plated at 30,000 cells per well in a 24-well plate with Ab/Am-free DMEM/10% FBS. The following day, cells were washed twice with PBS and the PBS replaced with OptiMEM. Cells were transfected with 50 nM siRNA using Lipofectamine RNAiMax (13778150; Life Technologies). At 16 h post-siRNA transfection, cells were washed twice with PBS and the PBS was replaced with Ab/Am-free DMEM/10% FBS. Cells were transfected with pGL3

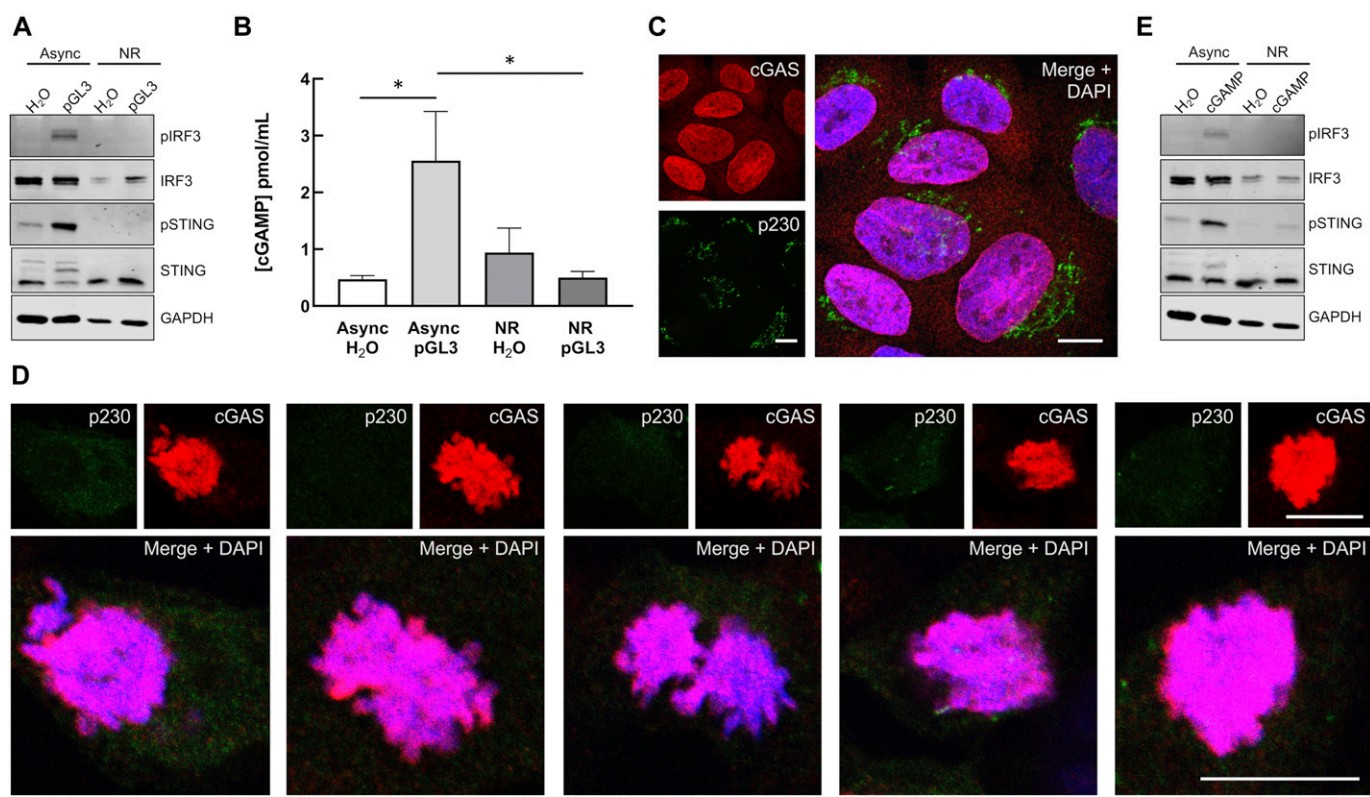

**Figure 6. cGAS and STING are nonresponsive during mitosis.**
**(A)** Mitotic phospho-IRF3 and phospho-STING responses to pGL3. Asynchronous and arrested cells were stimulated by transfection of 500 ng pGL3 or water for 90 min before Western blots. **(B)** Mitotic cGAMP responses to pGL3. Asynchronous and arrested cells were transfected with 500 ng pGL3. Non-internalized transfection complexes were removed by media change 2 h after the initial transfection. 5 h post-transfection, cGAMP levels were measured by ELISA. *$P < 0.05$, n = 2 biological replicates, with n = 2 technical replicates each. **(C, D)** cGAS subcellular localization and Golgi morphology in asynchronous (C) and arrested (D) HaCaT cells. Scale bars = 10 $\mu$m. **(E)** Mitotic phospho-IRF3 and phospho-STING responses to exogenous cGAMP. Asynchronous and arrested cells were treated with 25 $\mu$g/ml 2′-3′cGAMP for 2 h before Western blots. Source data are available for this figure.

24 h post-siRNA transfection, as described above. At 90 min post-transfection, samples were collected for Western blotting as described above.

### Golgi disruption

HaCaTs were plated at 60,000 cells per well on glass coverslips in 24-well plates. The following day, cells were treated with 5 $\mu$M nocodazole (sc-3518; Santa Cruz), 10 $\mu$M golgicide A (sc-215103; Santa Cruz), or 150 nM brefeldin A (B6542; Sigma-Aldrich), or DMSO vehicle for 1 h before further experimental treatments, and drugs were left on for the duration of these experiments.

### Cell synchronization

HaCaTs were plated at 7.5 million cells on a 10-cm dish in 10% FBS/DMEM and allowed to reach 100% confluence. The day after plating, the medium was changed to 1% FBS/DMEM and cells remained under contact inhibition at low-serum conditions for 48 h to synchronize at G₀. Cells were then replated at 30,000 cells per well on 24-well plates with or without glass coverslips in 10% FBS/DMEM and allowed to progress through the S phase. After 24 h, the cells were treated with 50 ng/ml nocodazole (sc-3518; Santa Cruz) for 12 h to synchronize at

the prometaphase, at which point they were released with two PBS washes and incubated in 10% FBS/DMEM. Cells were transfected with pGL3 as described above, and either harvested for Western blotting or prepared for IF or cell cycle analysis as described.

### Cell cycle analysis

Cell cycle status was analyzed by PI incorporation and flow cytometry. HaCaTs synchronized in the manner described above. At various time points during the synchronization and release from the prometaphase, cells were collected by trypsinization and pelleted at 500$g$ for 10 min at 4°C. The pellet was resuspended in ice-cold 70% ethanol to fix the cells and stored at −20°C until ready for PI staining. Fixed cells were pelleted at 1,000$g$ for 15 min at 4°C, resuspended in PBS, pH 7.4, containing 40 $\mu$g/ml PI and 500 $\mu$g/ml RNase A, and incubated at 37°C for 30 min. PI-stained cells were analyzed using the BD Biosciences FACSCanto-II flow cytometer and Diva 8.0 software.

### Immunofluorescent staining

HaCaTs were plated at 60,000 cells per well on glass coverslips in 24-well plates. The following day, cells were treated and transfected as described above. For mitotic sync experiments: HaCaTs were synchronized

to prometaphase as described above. For all experiments, after transfection, cells were fixed with 2% paraformaldehyde/PBS for 10 min at RT and permeabilized with 0.2% Triton X-100/PBS for 10 min at RT. Samples were blocked in 4% BSA/1% goat serum/PBS overnight at 4°C. Rabbit polyclonal anti-TGN46 (T7576. 1:500; Sigma-Aldrich), mouse monoclonal anti-p230 (611280, 1:500; BD Biosciences), mouse monoclonal anti-IRF3 (ab50772, 1:100; Abcam), rabbit anti-cGAS (15102, 1:100; Cell Signaling), and rabbit monoclonal anti-STING (ab181125, 1:500; Abcam) were used as primary antibodies. Alexa Fluor–488, Alexa Fluor–555, and Alexa Fluor–647 labeled goat antimouse and goat antirabbit secondary antibodies (A11029, A21424, A21429, and A21236; Life Technologies) were used at 1:1,000. Samples were then stained with 4′,6-diamidino-2-phenylindole (DAPI) (D9542-10MG; Sigma-Aldrich) at 1 µg/ml for 30 s. Coverslips were mounted on glass slides with Prolong Antifade Diamond (P36970; Life Technologies) and analyzed by confocal microscopy.

## Confocal microscopy

After the preparation of IF slides, confocal microscopy was performed using a Zeiss LSM880 system with 405, 488, and 543 nm lasers. Samples were examined using an oiled 63× objective, and Z-stacks with a 0.32 µm depth per plane were taken of each image. Representative single-plane images and Z-stacks were processed with the Zen Blue software.

## Colocalization analysis

Manders' overlap coefficients ([102]) for a STING:p230 and IRF3:DAPI channels within individual Z-stacks were determined using the JACoP plugin ([103]) on ImageJ ([104]). Manual thresholds were set below saturation. Individual Manders overlap coefficient values and mean values from multiple Z-stacks (each containing multiple cells), across multiple fields of view, were plotted with GraphPad Prism software.

## cGAMP stimulation

HaCaTs were plated at 60,000 asynchronous cells or 130,000 synchronous cells per well in 24-well plates. Cells were treated with 12.5 or 25 µg/ml 2′-3′cGAMP (tlrl-nacga23; Invivogen) for 2 h as indicated, then harvested for Western blotting as described above.

## SDS–PAGE and Western blotting

Samples were resolved by SDS–PAGE and transferred onto a 0.45-µm nitrocellulose membrane. Rabbit monoclonal anti-GAPDH (2118, 1:5,000; Cell Signaling), mouse monoclonal anti-IRF3 (50772, 1:100; Abcam), rabbit monoclonal anti-TBK1 (3504, 1:1,000; Cell Signaling), and rabbit monoclonal anti-STING (13647, 1:1,000; Cell Signaling) blots were blocked in 5% nonfat powdered milk dissolved in Tris-buffered saline containing 0.1% Tween (TBST). Rabbit monoclonal anti–phospho-IRF3 (Ser396 4947, 1:1,000; Cell Signaling), rabbit monoclonal anti–phospho-STING (Ser366 19781, 1:1,000; Cell Signaling), rabbit monoclonal anti–phospho-TBK1 (Ser172 5483, 1:1,000; Cell Signaling), and rabbit monoclonal anti-phospho-H3 (Ser10 3377, 1:10,000; Cell Signaling) blots were blocked in 100% Odyssey Blocking Buffer (927–40000; LI-COR). Goat antirabbit DyLight 680

(3568; Pierce), goat antimouse DyLight 680 (35518; Pierce), goat antirabbit DyLight 800 (535571; Pierce), and goat antimouse DyLight 800 (35521; Pierce) were used as secondary antibodies at 1:10,000 in either 50% Odyssey Blocking Buffer/TBST or 5% milk/TBST. Blots were imaged with the Licor Odyssey Infrared Imaging System. Band intensities were quantified by densitometry using ImageJ v1.52a ([104]).

## cGAMP ELISA

After Golgi vesiculation and mitotic sync experiments, cells were washed 1× with PBS and prepared for cGAMP quantification by the 2′,3′-Cyclic GAMP Direct ELISA Kit (K067-H1; Arbor Assays), following the manufacturer's protocol. cGAMP concentrations were normalized by total protein concentration, as determined by BCA Assay (23225; Thermo Fisher Scientific).

## RT-qPCR experiments

After Golgi disruption with GCA and transfection with pGL3, total RNA was prepared from HaCaTs using the QIAGEN RNeasy Mini Kit (74104; QIAGEN). RNA was isolated at 4 and 8 h post-pGL3 transfection. In the final step, RNA was eluted in 45 µl RNase/DNase-free water. RNA samples were purified using the TURBO DNA-free Kit (AM1907; Life Technologies), and cDNA was prepared using the High Capacity cDNA Reverse Transcription Kit (4368814; Thermo Fisher Scientific). To prepare cDNA, 1 µg of RNA was used per 40 µl final reaction volume, yielding an estimated cDNA concentration of 25 ng/µl. cDNA was diluted to 3.3 ng/µl with RNase/DNase-free water before use in qPCR. Reactions with specific ISG/IFN or TATA-binding protein (TBP) housekeeper primers were prepared using the PowerUp SYBR Green Master Mix Kit (A25742; Thermo Fisher Scientific) and loaded onto a 384-well plate. The reactions were run on a QuantStudio6 Flex Real-Time PCR System (Thermo Fisher Scientific). Delta-delta-cycle threshold (ΔΔCt) was determined relative to vehicle treated samples. Viral RNA levels were normalized to TBP housekeeper and depicted as fold change over vehicle treated samples. Error bars indicate the SEM from n = 3 technical replicates. Primer sequences are as follows: TBP-for; 5′-TAAACTTGACCTAAA GACCATTGCA-3′, TBP-rev; 5′-CAGCAAACCGCTTGGGATTA-3′, IFNB1-for; 5′-CTTGGATTCCTACAAAGAAGCAGC-3′, IFNB1-rev; 5′-TCCTCCTTCTGGAACTGC TGCA-3′, viperin-for; 5′-CCAGTGCAACTACAAATGCGGC-3′, viperin-rev, 5′-CGGTCTTGAAGAAATGGCTCTCC-3′, IFI6-for; 5′-TGATGAGCTGGTCTG CGATCCT-3′, IFI6-rev; 5′-GTAGCCCATCAGGGCACCAATA-3′, HERC5-for; 5′-CAACTGGGAGAGCCTTGTGGTT-3′, HERC5-rev; 5′-CTGGACCAGTTTG CTGAAAGTGG-3′, IFIT2-for; 5′-GGAGCAGATTCTGAGGCTTTGC-3′, IFIT2-rev; 5′-GGATGAGGCTTCCAGACTCCAA-3′, IFIT3-for; 5′-CCTGGAATGCTTACGG-CAAGCT-3′, IFIT3-rev; 5′-GAGCATCTGAGAGTCTGCCCAA-3′, CXCL10-for; 5′-GGTGAGAAGAGATGTCTGAATCC-3′, CXCL10-rev; 5′-GTCCATCCTTG-GAAGCACTGCA-3′, CXCL11-for; 5′-AAGGACAACGATGCCTAAATCCC-3′, CXCL11-rev; 5′-CAGATGCCCTTTTCCAGGACTTC-3′.

## Statistics

Statistical analyses were performed using Prism 6 (GraphPad Software). Statistics for the pIRF3 blot densitometry from Golgi vesiculation experiments were determined by an unpaired t test with the Holm–Sidak correction. Statistics for the cGAMP ELISAs and

pIRF3 densitometry in Fig 5 were determined by ordinary one-way ANOVA with the Tukey correction for multiple comparisons. RT-qPCR data were analyzed by two-way ANOVA with Tukey's multiple comparison. Colocalization statistics were calculated using a two-sample unpaired *t* test as recommended for colocalization analysis (105). Error bars on graphs represent standard error of mean. Number of biological and technical replicates is noted in figure legends.

## Supplementary Information

## Acknowledgements

We are grateful to Jim DeCaprio for advice on cell synchronization, Koenraad Van Doorslaer, Robert Jackson, and David Williams for their assistance with RT-qPCR, helpful discussion, and critical reading of this manuscript and Joshua Uhlorn for discussion and advice on statistics. We thank Anne Cress for the HaCaT cell line. We thank the University of Arizona (UA) BIO5 Media Facility, Patty Jansma of the UA Imaging Core-Marley, and John Fitch of the UA Cancer Center/Arizona Research Laboratories Cytometry Core Facility. This work was supported by grant R01AI108751 from the National Institute for Allergy and Infectious Diseases, grant R01GM136853 from the National Institute for General Medical Sciences, and grant from the Sloan Scholars Mentoring Network of the Social Science Research Council with funds provided by the Alfred P Sloan Foundation. BL Uhlorn is a graduate student supported by the National Science Foundation Graduate Research Fellowship Grant DGE-1143953.

### Author Contributions

BL Uhlorn: conceptualization, formal analysis, investigation, visualization, methodology, and writing—original draft, review, and editing.
ER Gamez: investigation and writing—original draft.
S Li: formal analysis, investigation, and visualization.
SK Campos: conceptualization, formal analysis, supervision, funding acquisition, visualization, methodology, project administration, and writing—original draft, review, and editing.

### Conflict of Interest Statement

The authors declare that they have no conflict of interest.

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
