## [Reviewer comments · Life Science Alliance]

Life Science Alliance

Attenuation of cGAS/STING Activity During Mitosis

Brittany Uhlorn, Eduardo Gamez, Shuaizhi Li, and Samuel Campos

DOI: <https://doi.org/10.26508/lsa.201900636>

Corresponding author(s): Samuel Campos, University of Arizona, College of Medicine Tucson

Review Timeline:

Submission Date:	2019-12-26
Editorial Decision:	2020-01-23
Revision Received:	2020-06-18
Editorial Decision:	2020-07-07
Revision Received:	2020-07-08
Accepted:	2020-07-08

Transaction Report:

January 23, 2020

Re: Life Science Alliance manuscript #LSA-2019-00636

Dr. Samuel K Campos
College of Medicine Tucson Immunobiology
Immunobiology
1657 E. Helen Street
Keating Bldg Rm 429
Tucson, Arizona 85721-0240

Dear Dr. Campos,

Thank you for submitting your manuscript entitled "Attenuation of cGAS/STING Activity During Mitosis" to Life Science Alliance. The manuscript was assessed by expert reviewers, whose comments are appended to this letter.

As you will see, the reviewers think that your work is important. However, they also raise overlapping concerns, pointing out that some of your conclusions are not sufficiently supported at this stage. We would thus like to invite you to submit a revised version of your manuscript to us, addressing the reviewers concerns. Doing so seems overall straightforward, requiring however a few additional experiments. Please do get in touch in case you would like to discuss specific revision points further with us.

Thank you for this interesting contribution to Life Science Alliance. We are looking forward to receiving your revised manuscript.

Sincerely,

B. MANUSCRIPT ORGANIZATION AND FORMATTING:

Reviewer #1 (Comments to the Authors (Required)):

In the work by Uhlorn BL et al, the authors report that vesiculation of the Golgi apparatus during mitosis attenuates cGAS-STING pathway activity. Impairment of the cGAS-STING axis may therefore provide an additional safeguard mechanism to restrict potentially harmful self-DNA sensing responses during cell division. The authors demonstrate a loss of function of the cGAS-

STING pathway in artificially induced Golgi vesiculated cells. The same observation was made in starved and pharmacologically synchronized cells where cGAS/STING activity was reduced in Golgi dispersed mitotic cells.

This study represents a relevant observation in the field of antiviral innate immunology and is crucial for the comprehension of the tight regulation of the cGAS/STING pathway during mitosis. The manuscript represents a nicely performed set of experiments. The total body of experimental data is convincing but looks somewhat underdeveloped at this stage and critically lacks some mechanistic insight. The article is logical in presentation and properly-written. Methodology is correctly described but some references acknowledging previous findings from other groups referring to some of the author's findings are missing.

Overall, the manuscript by Uhlorn et al is of interest for the broad readership of Life Science Alliances, but additional experiments have to be performed to increase the mechanistic deepness of the observations made by the authors. Listed below are some major and more minor comments that the authors should address in a revised version of their manuscript.

- The figure 1 does not add much novelty to the existing literature. Indeed, several groups already reported the capacity of HaCat cells to mount a potent cGAS/STING-dependent antiviral response to foreign DNA or cGAMP. These studies including among others (Almine JF, Nat. Comm. 2017; Dunphy G, Mol Cell, 2018; Olnagier D, Nat Comm, 2018; Skouboe MK, PLoS Pathogens, 2018, etc...) should be appropriately referenced.
- In figure 2, the authors use Golgicide A (GCA) as an artificial inducer of Golgi vesiculation. This compound has been previously reported in Gui et al, Nature, 2019 where the compound was demonstrated to alter STING trafficking to the ERGIC. This reference should be included where appropriate in the text. Both GCA and brefeldin A (BFA) inhibits Arf1, since BFA has also been reported to inhibit STING trafficking and signaling, can the authors also make the same link with Golgi vesiculation and STING inhibition in presence of BFA?
- Also related to Figure 2 and the use of Golgicide A, it would be nice for the authors to include more functional readouts of the type I IFN response impairment in GCA-treated cells such as qPCR of WB validation of antiviral genes/measurement of type I IFN release in response to DNA.
- In figure 2, quantification of STING expression in response to GCA is needed to address whether the compound alone affects STING expression or not? Also, from the confocal images, it looks like the Golgi network is not only dispersed but is also much less express after the GCA treatment. Can the authors comment on that and possibly find a way to illustrate that GCA only lead to a Golgi dispersion in the cell rather than a crude alteration of this biological compartment? Panel E and F are not included in the legend of Figure 2.
- In figure 3, despite the nice protocol that the authors have developed to synchronize their HaCat cells, it would be ideal for them to recapitulate the findings by selectively silencing/knocking-out or knocking-in some of the proteins involved in cell division to recapitulate the findings observed after starvation and nocodazole treatment. Same comment applies to figure 2. If authors can find a way to genetically alter some of the proteins involved in Golgi vesiculation, that would definitely strengthen the conclusions made with the Golgicide A.
- The major point that the authors need to address is the identification of the mechanistic link between Golgi vesiculation and impairment of STING signaling. Does Golgi vesiculation only leads to

an impairment of STING trafficking to the ERGIC/ER which further explains the absence of downstream signaling? Is there any interaction of STING with TBK1 and/or IRF3 in Golgi vesiculated/mitotic cells?

- In general, the authors need to provide quantifications of their different stainings to further strengthen the accuracy of their confocal data.

Reviewer #2 (Comments to the Authors (Required)):

By essentially two experimental settings (artificial Golgi vesiculation with a GBF-inhibitor in interphase cells and mitotic arrest with nocodazole), the authors found that Golgi integrity is critical for STING activation. To reveal the molecular mechanism underlying cGAS/STING activation is one of the hot topics in life sciences, and therefore this study is very timely.

The experiments are well designed and the data are essentially convincing. However, there are several concerns that have to be cleared before its publication.

Major critiques:

(1) I suggest to perform cGAMP stimulation in the experiments shown in Figure 2, as in Figure 4.

(2)

* The fluorescent images of STING staining in Figure 2 are very dim. Brighter images? If the expression levels of endogenous STING in HaCaTs is not enough for the indirect immunofluorescence detection of STING, fluorescent-tagged STING, such as EGFP-STING, may be useful.

* Magnified images should be shown.

* p230 is a peripherally membrane-associated protein and I wonder if the Golgi localization of p230 will be affected by GCA treatment. TGN46, a transmembrane protein in the TGN, may be useful to mark the Golgi.

Minor critiques:

(3) Related to Figure 2, does STING reach the Golgi in the presence of GCA?

(4) The authors would like to examine the phosphorylated TBK1 (the active form of TBK1) in western blot. This experiment is not mandatory, but examining pTBK1 is a kind of routine in this field.

(5) several typos:

page 3 line 10 in the right column

"quiescient" should read "quiescent"

page 3 line 11 in the right column

"reentry" should read "re-entry"

page 3 line 20 in the right column

"NOC-synched cells" should read "NOC-synchronized cells"

Reviewer #3 (Comments to the Authors (Required)):

The article entitled "Attenuation of cGAS/STING Activity During Mitosis" by Uhlorn et al. describe the control of cGAS signaling during cell division. As the nuclear membrane is no longer intact during mitosis, how cells prevent cGAS from recognizing self-DNA is important topic, and it has implications for understanding the development of autoimmunity. The authors report that interaction of cGAS with mitotic chromosomes leads to only a minimal activation. On the other hand, structural disruption of Golgi that occurs during mitosis has a significant impact on STING activation of IRF3 without affecting cGAS production of cGAMP itself. Overall, this manuscript addresses an important research question. Please see below for specific comments.

The cGAS signaling in HaCaT's cell line should thoroughly be characterized. The authors should check TBK1 phosphorylation as well as IFN α and/or IFN β expression (at the mRNA and/or protein level).

The authors should use additional DNA ligands in addition to the plasmid DNA used.

Fig 2. If golgicide A affects the activation of TBK1 and the expression of IFN α and/or IFN β should also be tested.

The experiments in Fig 3 should include appropriate control stimulations (such as polyI:C, etc.). Biochemical demonstration of the effect of mitosis on STING recruitment to Golgi would strengthen the findings.

The figure legends should also indicate if the bar graphs represent technical or biological replicates (and the number of repeats).

Thank you to the reviewers for your thoughtful critiques and suggestions. We have revised the manuscript to address all the concerns we could, and we have reformatted it for ease of review (now double spaced with line numbers). Specific responses to your comments are below.

Reviewer #1 (Comments to the Authors (Required)):

In the work by Uhlorn BL et al, the authors report that vesiculation of the Golgi apparatus during mitosis attenuates cGAS-STING pathway activity. Impairment of the cGAS-STING axis may therefore provide an additional safeguard mechanism to restrict potentially harmful self-DNA sensing responses during cell division. The authors demonstrate a loss of function of the cGAS-STING pathway in artificially induced Golgi vesiculated cells. The same observation was made in starved and pharmacologically synchronized cells where cGAS/STING activity was reduced in Golgi dispersed mitotic cells.

This study represents a relevant observation in the field of antiviral innate immunology and is crucial for the comprehension of the tight regulation of the cGAS/STING pathway during mitosis. The manuscript represents a nicely performed set of experiments. The total body of experimental data is convincing but looks somewhat underdeveloped at this stage and critically lacks some mechanistic insight. The article is logical in presentation and properly-written. Methodology is correctly described but some references acknowledging previous findings from other groups referring to some of the author's findings are missing.

Overall, the manuscript by Uhlorn et al is of interest for the broad readership of Life Science Alliances, but additional experiments have to be performed to increase the mechanistic deepness of the observations made by the authors. Listed below are some major and more minor comments that the authors should address in a revised version of their manuscript.

- The figure 1 does not add much novelty to the existing literature. Indeed, several groups already reported the capacity of HaCat cells to mount a potent cGAS/STING-dependent antiviral response to foreign DNA or cGAMP. These studies including among others (Almine JF, Nat. Comm. 2017; Dunphy G, Mol Cell, 2018; Olagnier D, Nat Comm, 2018; Skouboe MK, PLoS Pathogens, 2018, etc...) should be appropriately referenced.

We sincerely apologize for missing these papers that used HaCaT cells for studies on cGAS/STING. The cGAS/STING field has rapidly expanded but we have now referenced these, and other papers using HaCaT cells (see lines 72-73). We agree figure 1 does not add much novelty alone, but feel it is still important to include as cell lines can vary in their behavior across different labs. Furthermore, it does show that the pIRF3 response is quite dependent on cGAS/STING in these cells.

- In figure 2, the authors use Golgicide A (GCA) as an artificial inducer of Golgi vesiculation. This compound has been previously reported in Gui et al, Nature, 2019 where the compound was demonstrated to alter STING trafficking to the ERGIC. This reference should be included where appropriate in the text. Both GCA and brefeldin A (BFA) inhibits Arf1, since BFA has also been reported to inhibit STING trafficking and signaling, can the authors also make the same link with Golgi vesiculation and STING inhibition in presence of BFA?

Thank you for pointing this out, we have referenced Gui et al. and we have included additional experiments with BFA and high-dose nocodazole which will fragment but not vesiculate the Golgi (see new figures 2 and 3 and lines 89-106). BFA and GCA behaved identically in these assays. Interestingly fragmentation of the Golgi by nocodazole did not significantly affect cGAS/STING activity, suggesting the factors necessary for STING activation are retained on these fragmented Golgi structures.

- Also related to Figure 2 and the use of Golgicide A, it would be nice for the authors to include more functional readouts of the type I IFN response impairment in GCA-treated cells such as qPCR of WB validation of antiviral genes/measurement of type I IFN release in response to DNA.

We have now included RT-qPCR analysis of downstream IFN/ISG/cytokine responses (see new figure 4 and lines 153-158). GCA blunted all these DNA-dependent responses.

- In figure 2, quantification of STING expression in response to GCA is needed to address whether the compound alone affects STING expression or not? Also, from the confocal images, it looks like the Golgi network is not only dispersed but is also much less express after the GCA treatment. Can the authors comment on that and possibly find a way to illustrate that GCA only lead to a Golgi dispersion in the cell rather than a crude alteration of this biological compartment? Panel E and F are not included in the legend of Figure 2.

We now included STING in the GCA western blots (see new figure 3). Band intensity does not suggest lower expression of STING upon GCA treatment. Differences in IF staining intensity are likely due to differences in cell thickness and subcellular distribution of STING rather than protein levels. Apologies for the panel labeling mistakes, these have been corrected.

- In figure 3, despite the nice protocol that the authors have developed to synchronize their HaCat cells, it would be ideal for them to recapitulate the findings by selectively silencing/knocking-out or knocking-in some of the proteins involved in cell division to recapitulate the findings observed after starvation and nocodazole treatment. Same comment applies to figure 2. If authors can find a way to genetically alter some of the proteins involved in Golgi vesiculation, that would definitely strengthen the conclusions made with the Golgicide A.

Unfortunately silencing or knockout of genes involved in Golgi dispersal/integrity interfere with the G2/M checkpoint and cell division/proliferation, and preclude use in timecourse experiments where the Golgi must be dispersed just prior to stimulation of cGAS/STING. Likewise, these genetic approaches lack the temporal control required for synchronization/release experiments to analyze a population of cells as they progress through mitosis. It is also challenging to ensure that all the HaCaT cells in a given population adequately take up the reagents required for efficient genetic silencing.

- The major point that the authors need to address is the identification of the mechanistic link between Golgi vesiculation and impairment of STING signaling. Does Golgi vesiculation only leads to an impairment of STING trafficking to the ERGIC/ER which further explains the absence of downstream signaling? Is there any interaction of STING with TBK1 and/or IRF3 in Golgi vesiculated/mitotic cells?

Unfortunately we were unable to adequately and reproducibly generate good data using IP methods against STING. According to our data, STING is blocked from trafficking and activation (pSTING) by Golgi dispersal. Data from others suggests that until STING reaches the Golgi it is unable to efficiently recruit TBK1 (to generate pSTING), oligomerize, and recruit IRF3 (to generate pIRF3). The exact mechanisms for activation/trafficking of STING remain to be determined. Further work will be necessary to understand the basis for Golgi-dependent STING activation.

- In general, the authors need to provide quantifications of their different stainings to further strengthen the accuracy of their confocal data.

We have now quantified our colocalization experiments with various Golgi-perturbing compounds (see new figure 2). We have also added arrowheads to indicate overlap in the micrographs.

Reviewer #2 (Comments to the Authors (Required)):

By essentially two experimental settings (artificial Golgi vesiculation with a GBF-inhibitor in interphase cells and mitotic arrest with nocodazole), the authors found that Golgi integrity is critical for STING activation. To reveal the molecular mechanism underlying cGAS/STING activation is one of the hot topics in life sciences, and therefore this study is very timely.

The experiments are well designed and the data are essentially convincing. However, there are several concerns that have to be cleared before its publication.

Major critiques:

- (1) I suggest to perform cGAMP stimulation in the experiments shown in Figure 2, as in Figure 4.

We have now added data with cGAMP stimulation in our GCA experiments (see new figure 3E). Results agree with the other findings.

(2)

* The fluorescent images of STING staining in Figure 2 are very dim. Brighter images? If the expression levels of endogenous STING in HaCaTs is not enough for the indirect immuno-fluorescence detection of STING, fluorescent-tagged STING, such as EGFP-STING, may be useful.

* Magnified images should be shown.

* p230 is a peripherally membrane-associated protein and I wonder if the Golgi localization of p230 will be affected by GCA treatment. TGN46, a transmembrane protein in the TGN, may be useful to mark the Golgi.

We have intentionally avoided ectopic expression of STING or EGFP STING as we have found it leads to auto-activation. Rather, we have adjusted the image settings across all the micrographs to better illustrate what is seen by eye on the scope. We use both TGN46 and p230 to mark the Golgi/trans-Golgi in the IF experiments in new figure 2.

Minor critiques:

(3) Related to Figure 2, does STING reach the Golgi in the presence of GCA?

STING appears to remain punctate, likely at ERES, according to the literature on GCA

(4) The authors would like to examine the phosphorylated TBK1 (the active form of TBK1) in western blot. This experiment is not mandatory, but examining pTBK1 is a kind of routine in this field.

We have now included TBK1/pTBK1 blots in our GCA experiments in new figure 2. Although differences are not as stark as STING/pSTING, the data do show increased levels of pTBK1 when expected.

(5) several typos:

page 3 line 10 in the right column

"quiescient" should read "quiescent"

page 3 line 11 in the right column

"reentry" should read "re-entry"

page 3 line 20 in the right column

"NOC-synched cells" should read "NOC-synchronized cells"

Thanks for finding these mistakes, we have now corrected them

Reviewer #3 (Comments to the Authors (Required)):

The article entitled "Attenuation of cGAS/STING Activity During Mitosis" by Uhlorn et al. describe the control of cGAS signaling during cell division. As the nuclear membrane is

no longer intact during mitosis, how cells prevent cGAS from recognizing self-DNA is important topic, and it has implications for understanding the development of autoimmunity. The authors report that interaction of cGAS with mitotic chromosomes leads to only a minimal activation. On the other hand, structural disruption of Golgi that occurs during mitosis has a significant impact on STING activation of IRF3 without affecting cGAS production of cGAMP itself. Overall, this manuscript addresses an important research question. Please see below for specific comments.

The cGAS signaling in HaCaTs cell line should thoroughly be characterized. The authors should check TBK1 phosphorylation as well as IFN α and/or IFN β expression (at the mRNA and/or protein level).

We have now added TBK1 data (new figure 2) and have looked at IFN/ISG/cytokine transcripts (new figure 4)

The authors should use additional DNA ligands in addition to the plasmid DNA used.

In addition to pGL3 plasmid, we have also now included a short 60-mer dsDNA oligo (HSV-60) and HMW calf thymus DNA (new figure 3). The short oligo was less efficient at activating cGAS/STING, consistent with a length requirement for activation.

Fig 2. If golgicide A affects the activation of TBK1 and the expression of IFN α and/or IFN β should also be tested.

We have now included RT-qPCR of IFN/ISG/cytokine transcripts in response to DNA +/- GCA (new figure 4)

The experiments in Fig 3 should include appropriate control stimulations (such as polyI:C, etc.).

We have decided only to focus on the cGAS/STING pathway for the mitosis experiments in this paper.

Biochemical demonstration of the effect of mitosis on STING recruitment to Golgi would strengthen the findings.

We agree, but unfortunately we have not been able to gather strong and reproducible data using IP approaches. We have added additional blots indicating that STING activation (pSTING) depends on the Golgi.

The figure legends should also indicate if the bar graphs represent technical or biological replicates (and the number of repeats).

We have added this info, apologies for the oversight.

Thank you all for the constructive critiques and suggestions. We feel the revised manuscript is now stronger and look forward to further work towards understanding the mechanisms for Golgi-dependent activation of STING.

July 7, 2020

RE: Life Science Alliance Manuscript #LSA-2019-00636RR

Dr. Samuel K Campos
University of Arizona, College of Medicine Tucson
Department of Immunobiology
1657 E. Helen Street
Keating Bldg Rm 429
Tucson, Arizona 85721-0240

Dear Dr. Campos,

Thank you for submitting your revised manuscript entitled "Attenuation of cGAS/STING Activity During Mitosis". It has now been seen by two of the original referees. As you can see, the referees find that the study is significantly improved during revision and recommend publication. We would be happy to publish your paper in Life Science Alliance pending final revisions necessary to meet our formatting guidelines.

- please look at our Manuscript Preparation guidelines and separate your manuscript sections accordingly
- please add the author contributions to the main manuscript
- please list 10 authors et al. in the references
- Please revise Figure Legend 2 mentioning panels in alphabetical order
- please remove subpanel specifiers A-H from Figure 4

A. FINAL FILES:

B. MANUSCRIPT ORGANIZATION AND FORMATTING:

Sincerely,

Reilly Lorenz
Editorial Office Life Science Alliance
Meyerhofstr. 1
69117 Heidelberg, Germany
t +49 6221 8891 414
e contact@life-science-alliance.org
www.life-science-alliance.org

Reviewer #2 (Comments to the Authors (Required)):

The authors adequately addressed to Reviewer #2's concerns. I believe that this Ms is ready for publication.

Reviewer #3 (Comments to the Authors (Required)):

The revised version has adequately addressed the concerns raised. The findings of this manuscript would be valuable to the innate immunity field.

July 8, 2020

RE: Life Science Alliance Manuscript #LSA-2019-00636RRR

Dr. Samuel K Campos
University of Arizona, College of Medicine Tucson
Department of Immunobiology
1657 E. Helen Street
Keating Bldg Rm 429
Tucson, Arizona 85721-0240

Dear Dr. Campos,

Thank you for submitting your Research Article entitled "Attenuation of cGAS/STING Activity During Mitosis". It is a pleasure to let you know that your manuscript is now accepted for publication in Life Science Alliance. Congratulations on this interesting work.

DISTRIBUTION OF MATERIALS:

You can contact the journal office with any questions, contact life-science-alliance.org

Again, congratulations on a very nice paper. I hope you found the review process to be constructive and are pleased with how the manuscript was handled editorially. We look forward to future exciting submissions from your lab.

Sincerely,

Reilly Lorenz
Editorial Office Life Science Alliance
Meyerhofstr. 1
69117 Heidelberg, Germany
t +49 6221 8891 414
e contact@life-science-alliance.org
www.life-science-alliance.org